# $\alpha$-RANK: UNIFIED ITEM-FAIR RANKING FROM A CO-OPERATIVE GAME THEORY VIEW

## ABSTRACT

Driven by economic and systematic considerations, the pursuit of item fairness in ranking has emerged as a prominent topic in recommendation and advertising applications. Prior research has suggested various fairness aspects can be aligned with the concept of distributive justice in sociology, such as utilitarianism, dealism, and egalitarianism. However, they fail to distinguish the distinctions and relationships among these fairness dimensions in ranking. In fact, item fairness can be viewed as a unified challenge of fairly allocating the constrained and fluctuating resources, from the perspective of cooperative game theory. In our work, we introduce the smooth $\alpha$-fairness objective for different fairness and unify item fairness as a cooperative game problem. In such games, items are considered as the players dividing the "cake" of user attention. In such games, we analyze the $\alpha$-fairness objective from a theoretical way and introduce an efficient approach called *α-rank*. Firstly, we re-form several important axioms in cooperative games to tell us how item fairness principles exhibit when the resource "cake" changes in ranking. Then we designed *α-rank*, which applies the optimal transport to conduct item fairness. Theoretical analysis provides an upper bound, showcasing the maximum total utility loss across different fairness degrees. we conducted experiments in two ranking applications: recommendation and advertising. The experimental results demonstrate that *α-rank* effectively and efficiently outperforms the baseline methods.

## 1 INTRODUCTION

Ranking techniques have found extensive application in web-based platforms, such as determining which items to display to users in recommendation and advertising scenarios with limited exposure slots (Xu et al., 2018; Baeza-Yates et al., 1999). Recently, researchers have emphasized the importance of item fairness in ranking, as it not only prevents monopolization but also contributes to the creation of a healthier ecosystem (Xu et al., 2023a; Patro et al., 2020; Do et al., 2021; Li et al., 2022; Lipani, 2016).

Different from the user fairness, which pertains to ensuring that everyone has fundamental rights and responsibilities (Matsumoto & Juang, 2016; Abdollahpouri et al., 2019). Item fairness, which aims to equitably distribute items among users, is closely aligned with the concept of distributive justice (Lamont, 2017; Matsumoto & Juang, 2016) in sociology. Previous research papers advocate item from distinct principles: utilitarianism objective (Baeza-Yates et al., 1999; Lacerda et al., 2006), which focused on maximizing summarization of all item utilities; dealism objective aligned with proportion fairness (Ben-Porat & Tennenholtz, 2018), which aims to strive to achieve an allocation where items possess resources in proportion to their respective weights or importance; dgalitarianism objective aligned with max-min fairness (Xu et al., 2023a; Do et al., 2021; Patro et al., 2020), which equalizes the utilities of all item utilities involved in the decision-making process.

Although previous ranking models have introduced effective algorithms aligned with specific fairness aspects, they often lack a clear distinction between these underlying fairness principles. Inspired from cooperative game theory, the concept of item fairness in ranking closely resembles the notion of fair resource allocation (Matsumoto & Juang, 2016; Xu et al., 2023a), which primarily focuses on finding a suitable resource allocation method that caters to the utility of all involved parties

in an economic way. In a simple way, item fairness can be seen as a challenge involving resource allocation in scenarios where resources are both limited and subject to fluctuation.

In our work, we approach the issue of item fairness in ranking from a unified perspective rooted in cooperative game theory (Branzei et al., 2008; Peleg & Sudhölter, 2007). Within the framework of cooperative games, each item is viewed as a participant tasked with fairly dividing the "cake" of limited exposure slots. Inspired by cooperative games, we introduce the concept of $\alpha$-fairness (Xu & Cumanan, 2017; Bertsimas et al., 2012) to achieve a well-balanced equilibrium among item fairness principles. As $\alpha$ approaches 0, 1, and $\infty$, it corresponds to the utilitarianism, dealism, and egalitarianism solutions, respectively. Optimizing $\alpha$-fairness offers a smooth and adaptable approach to achieve item fairness in accordance with varying requirements.

Then we analyze the $\alpha$-fairness objective from a theoretical way and introduce an efficient approach called $\alpha$-ranking. Specifically, we begin by reforming several key axioms of cooperative games designed for item fairness. The axioms describe how different item fairness principles behave when the amount of resources, such as limited exposure slots, changes.

After that, we propose an efficient approach named $\alpha$-rank to efficiently tackle the $\alpha$-fairness optimization objective in ranking. Firstly, we identify an upper-bound function for the target problem, which conforms to the structure of a standard cooperative game and can be efficiently resolved. Then, we utilize Sinkhorn algorithm (Swanson et al., 2020) of optimal transport (OT) (Pham et al., 2020; Peyré et al., 2019) to map the upper-bound function back to the original space, thus arriving at our ranking results efficiently. Finally, we offer theoretical insights into the maximum loss of total item utilities under various $\alpha$ values through the upper-bound function.

We also apply $\alpha$-ranking into real-world ranking scenarios, specifically in recommendation and advertising, using two extensive public datasets. Experiment results demonstrate that $\alpha$-rank can achieve better performance while maintaining the efficiency required for industrial ranking systems.

## 2 RELATED WORKS

**Fairness principle:** Cultural perspectives on fairness exhibit significant variations, as extensively explored in sociological research (Tyler & Allan Lind, 2002; Tyler & Smith, 1995). In practice, two common fairness definitions are rooted: equality and equity (Matsumoto & Juang, 2016). Equality is defined as: *everyone is treated the same and provided the same resources to succeed*, which aims to ensure the fundamental rights and responsibilities of each individual. While equity is defined as: *ensuring that resources are equally distributed based on needs*, which is close to the concept of distributive justice (Lamont, 2017). In distributive justice, there are three types of allocation principles. Utilitarianism proposed by Aristotle (Sen, 1979), which aims to maximize the summation of utilities. As for the dealism proposed by Nash (Nash Jr, 1950), it focuses on reaching an agreement point based on the deals previously made by each side. Egalitarianism (Rawls, 1971) aims to equalize the utilities of all individuals. Item fairness is more related to the distributive justice realm. In this paper, we apply the cooperative games to unify the three principles in item fairness.

**Item fairness methods:** Regarding item fairness, previous work often focused on two types: individual fairness (Marras et al., 2022; Li et al., 2021), which concentrates on equitable treatment for individuals, and group fairness, which categorizes items groups (Ge et al., 2021; Xu et al., 2023a). Our work primarily focuses on individual fairness as the main objective, while group fairness can be formulated in a similar manner. For different fairness aspects, Current mainstream ranking systems (Rendle et al., 2012; Xue et al., 2017; Yang et al., 2019) apply utilitarianism to optimize the summation of platform profit. Dealism often relates to the proportion fairness (Bertsimas et al., 2011). For example, Ben-Porat & Tennenholtz (2018); Patro et al. (2020); Biswas et al. (2021) proposed the Shapley algorithm to reach the point. Optimizing objective of Egalitarianism could be Gini Index (Do & Usunier, 2022), max-min fairness (Xu et al., 2023a; Do et al., 2021) and distance of different groups (Jiang et al., 2021). However, they often focused on one type of fairness and failed to distinguish the connections between different fairness principles.

**Cooperative games:** The field of game theory (Von Neumann & Morgenstern, 1947), is commonly divided into: cooperative games and non-cooperative games. Different from non-cooperative game (Nash, 1951), cooperative game involves players whose interests are neither completely opposed nor completely coincident, allowing them to communicate and collaborate. In cooperative game

Table 1: Detailed explanations of variable in item fairness

| Symbol | Value | Application | Explanation |
|--------|-------|-------------|-------------|
| $\boldsymbol{v}_i$ | $\boldsymbol{v}_i = \sum_u w_{u,i} \boldsymbol{x}_{u,i}$ | amortized ranking (Xu et al., 2023a; Biega et al., 2018) | the utility of the item $i$ within the user arrival times |
| $w_{u,i}$ | $w_{u,i} = 1$ | exposure-based fairness (Xu et al., 2023a; Patro et al., 2020) | charging according to one exposure of item/advertise $i$ to user $u$ |
| | $w_{u,i} = \text{ctr}_{u,i}$ | CTR-based fairness (Rendle et al., 2012; Xue et al., 2017) | charging according to the user $u$ clicked on the item $i$ once |
| | $w_{u,i} = \text{ctr}_{u,i} * \text{cvr}_{u,i}$ | CVR-based fairness (Yang et al., 2019; Liu et al., 2021) | charging according to the user $u$ conversioned on the item $i$ once |
| $\text{ctr}_{u,i}, \text{cvr}_{u,i}$ | CTR/CTR value | CTR/CVR billing | CTR/CVR value of user $u$ to item $i$ |
| $\gamma_i$ | $\gamma_i = \beta_i$ | recommendation (Rendle et al., 2012) | $\beta_i$ serves as the adjustment factor for each item |
| | $\gamma_i = \text{bid}_i * \beta_i$ | advertising (Yang et al., 2019; Liu et al., 2021) | $\text{bid}_i$ represents the bidding value of the advertiser. |

theory, (Shapley et al., 1953) proposed the concept of Sharpley value, offering an approach for fairly allocating in cooperative games. Another approach, proposed by (Nash Jr, 1950), is rooted in the concept of bargaining. To highlight the weight of importance of players' bargaining power, Nash (Nash Jr, 1950) introduced a generalized framework of bargaining. Kalai & Smorodinsky (1975) proposed a solution that focuses on the proportion to the ideal utility of each player. Based on this, Kalai (1977) proposed a max-min method to equalize the utility of all players involved. In our research, we approach item fairness as a problem of equitable resource allocation for items, drawing inspiration from the perspective of cooperative game theory.

## 3 PROBLEM FORMULATION

We first define some notations for the problem. For vector $\boldsymbol{x} \in \mathbb{R}^n$, let $\boldsymbol{x}_i$ denote the $i$-th element of the vector. For vector $\boldsymbol{x} \in \mathbb{R}^{n \times m}$, let $\boldsymbol{x}_{i,j}$ denote the element of $i$-th row and $j$-th column. $\boldsymbol{A}_i$ denote the $i$-th column vector of $\boldsymbol{A}$. $\boldsymbol{x} \geq \boldsymbol{y}$ denotes element $\boldsymbol{x}_i$ should be greater or equal to $\boldsymbol{y}_i, \forall i$.

In this section, we will formulate the item fairness in ranking into a constrained optimization problem. In the context of ranking, we define $\mathcal{U}$ representing the set of users, $\mathcal{I}$ representing the set of items. When a user $u \in \mathcal{U}$ interacts with the system, the number of retrieved items is typically limited and is often defined by a constant value denoted as $K$. For each user $u$, the decision vector $\boldsymbol{x}_u \in \{0,1\}^{|\mathcal{I}|}$, where $\boldsymbol{x}_{u,i} = 1$ denotes item $i$ should be recommended to user $u$, otherwise, $\boldsymbol{x}_{u,i} = 0$. The utilities of items can be represented as a vector $\boldsymbol{v} \in \mathbb{R}_+^{|\mathcal{I}|}$, where utility of certain item $\boldsymbol{v}_i$ relates to the decision vector $\boldsymbol{x}_u$.

Then, we can write the ranking problem into the following mathematical program in a general way:

$$\boldsymbol{v}^f = \arg\max_{\boldsymbol{v} \in \mathcal{D}} f(\boldsymbol{v}), \quad \mathcal{D} = \{\boldsymbol{v}(\boldsymbol{x}_u) | \mathbf{1}^\top \boldsymbol{x}_u = K, \forall u \in \mathcal{U}\}, \tag{1}$$

where $f(\cdot)$ represents the fairness optimization objective of ranking, which can vary depending on different objectives or proposals. To better understand the item fairness application, we give an illustrated example in Appendix E. Previous studies proposed distinct types of optimization objectives that correspond to different principles of fair resource allocation in terms of $f(\cdot)$:

(1) Utilitarianism (w/o fairness) (Rendle et al., 2012): $f(\boldsymbol{v}) = \sum_i \gamma_i \boldsymbol{v}_i$

(2) Dealism (proportion fairness) (Li et al., 2022): $f(\boldsymbol{v}) = \sum_i \gamma_i \log \boldsymbol{v}_i$

(3) Egalitarianism (max-min fairness) (Xu et al., 2023a): $f(\boldsymbol{v}) = \min_i \gamma_i \boldsymbol{v}_i$,

where the $\gamma_i$ is the weight of each item. In rankings, $\boldsymbol{v}_i$ and $\gamma_i$ have different forms, which are listed in the Table 1. In the Table 1, CTR and CVR is the abbreviation of click-through-rate (CTR), and conversion-through-rate (CVR) (Yang et al., 2019).

For various fairness principles within the objectives of ranking: utilitarianism (Matsumoto & Juang, 2016) strives to maximize the overall utilities of the items, seeking to optimize the collective benefit. Dealism (Bertsimas et al., 2011) aims to strive to allocate items that possess resources in proportion to their respective weights $\gamma_i$. Egalitarianism (Bertsimas et al., 2011) aims to equalize the utilities of items by enhancing the utility of the worst-off items, promoting fairness through improved distribution of benefits. Previous work also proposed to trade-off the different optimizing objectives for item fairness (Abdollahpouri & Burke, 2019; Abdollahpouri et al., 2020; Hao et al., 2021; Naghiaei et al., 2022).

In cooperative games, the $\alpha$-fairness (Bertsimas et al., 2012) provides a smooth way to unify the three types of fairness principles:

$$f(\boldsymbol{v}; \alpha) = \begin{cases} \sum_i \frac{\boldsymbol{v}_i^{1-\alpha}}{1-\alpha} & \text{if } \alpha > 0, \alpha \neq 1 \\ \sum_i \log(\boldsymbol{v}_i) & \text{if } \alpha = 1 \end{cases}, \quad W(\alpha) = \max_{\boldsymbol{v} \in \mathcal{D}} f(\boldsymbol{v}; \alpha). \qquad (2)$$

where $\alpha$ approaches 0, 1, and $\infty$, it corresponds to the utilitarianism, dealism, and egalitarianism solutions, respectively.

## 4 OUR FRAMEWORK

In this section, we will first introduce five axioms of the $\alpha$- fairness in ranking from the view of cooperative game in a theoretical way. Then, we present the $\alpha$-ranking to efficiently and effectively solve the item fairness in ranking.

### 4.1 AXIOMS IN ITEM FAIRNESS

In this section, we will re-form several axioms in cooperative game theory (Bertsimas et al., 2011) that one might seek in an item-fair ranking system. These axioms show that how item fairness principles will behave when there are changes in available resources.

**Axiom 1 (Pareto Optimality)** *The utilities of items $\boldsymbol{v}^f$ is Pareto optimal, that is, there does not exist another solution $\boldsymbol{v} \in \mathcal{U}$ so that utility vector $\boldsymbol{v} \geq \boldsymbol{v}^f$ and $\boldsymbol{v}^f \neq \boldsymbol{v}$.*

**Axiom 2 (Symmetry)** *When two items $i$ and $j$ possess equal weight and charging weight values, indicated by $\gamma_i = \gamma_j$, they are expected to yield the same utility outcome, denoted as $\boldsymbol{v}_i = \boldsymbol{v}_j$.*

**Axiom 3 (Affine Invariance)** *If we have an affine operator $A(\boldsymbol{v}_i) = c_i \boldsymbol{v}_i$, $c_i > 0$, then fair allocation under ranking is equal to the affine transformation of the fair allocation under the original system, i.e. $\arg\max_{\boldsymbol{v}} f(A(\boldsymbol{v})) = A(\arg\max_{\boldsymbol{v}} f(\boldsymbol{v}))$.*

**Axiom 4 (Independence of Irrelevant Alternatives)** *If $\mathcal{D}_1, \mathcal{D}_2$ are two utility feasible set such that $\mathcal{D}_1 \subset \mathcal{D}_2$ and $\arg\max_{\boldsymbol{v} \in \mathcal{D}_2} f(\boldsymbol{v}) \in \mathcal{D}_1$, then $\arg\max_{\boldsymbol{v} \in \mathcal{D}_1} f(\boldsymbol{v}) = \arg\max_{\boldsymbol{v} \in \mathcal{D}_2} f(\boldsymbol{v})$.*

**Axiom 5 (Resource Monotonicity)** *Let $\mathcal{D}_1, \mathcal{D}_2$ be two utility sets and $\mathcal{D}_1 \subset \mathcal{D}_2$ and $\mathcal{D}_1 \neq \mathcal{D}_2$, we have $\arg\max_{\boldsymbol{v} \in \mathcal{D}_1} f(\boldsymbol{v}) \leq \arg\max_{\boldsymbol{v} \in \mathcal{D}_2} f(\boldsymbol{v})$.*

Axiom 1 (Pareto optimality) ensures that no situation can arise where the utility of two items can simultaneously increase. Axiom 2 (Symmetry) ensures ranking model cannot differentiate the items by their attributes. Axiom 3 (Affine Invariance) guarantees that the ranking outcome remains unchanged regardless of the choice of utility numeraire. Axiom 4 (Independence of Irrelevant Alternatives) illustrates that if resources are decreased for an item and the original solution lies within the feasible region, then the solution remains the same as the original. Axiom 5 (Resource Monotonicity) illustrates that increasing the feasible set will give each item equal or greater utility. Detailed analysis can be seen in Axiom 1-4 (Nash Jr, 1950) and Axiom 5 (Kalai & Smorodinsky, 1975).

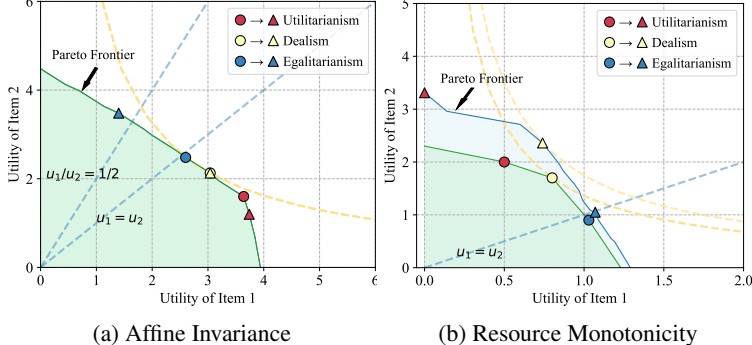

(a) Affine Invariance  (b) Resource Monotonicity

Figure 1: Toy examples to illustrate the axioms of item fairness. Two items (item 1 and item 2) are recommended to 30 users, with the constraint that each user can only be exposed to one item (i.e. $K = 1$). Circles and triangles are utilized to visually depict the shifts in optimal solutions for each fairness criterion when faced with changes in available resources.

**Theorem 1** *Utilitarianism, dealism, and egalitarianism all adhere to Axioms 1, 2, and 4. However, utilitarianism and dealism fail to meet Axiom 5 (Resource Monotonicity), while utilitarianism and egalitarianism do not conform to Axiom 3.*

**Remark 1** *The axioms indicate that as $\alpha$ varies from 0 to 1, the system disregards the numeraire of utilities and allocates resources in proportion to their weight $\gamma_i$ more often.*

**Remark 2** *The axioms suggest that as $\alpha$ increases from 1 to $\infty$, the system instructs the platform to enhance the utility of the worst-off item and simultaneously improve the utility of all items when the resource increases more often.*

The proof of Theorem 1 can be seen in Appendix A. The theorem indicates that various fairness principles exhibit varying performance as the available resources change.

In order to better understand the axioms from the view of cooperative games, we conducted a simulation to analysis of two axioms. Figure 1 illustrates a real ranking scenario where the resource "cake" to explore how different fairness principles perform in ranking tasks. We set each item's weight $\gamma_i$ to 2 and 3, respectively.

Figure 1 (a) illustrates that optimal points corresponding to various fairness principles if change the numeraire of resource "cake". In this simulation, we have doubled the utility of item 2 while keeping the utility of item 1 unchanged. The experimental findings demonstrate that utilitarianism and egalitarianism do not adhere to Axiom 3 (Affine Invariance). Conversely, dealism maintains its behavior of allocating resources in proportion to their weight $\gamma_i$ for items 2 and 3, respectively, regardless of the choice of resource numeraire.

Figure 1 (b) depicts the optimal points corresponding to various fairness principles if we increase the size of the resource "cake". In this illustration, the green regions represent the original feasible region, while the blue regions indicate the expanded region where additional resources are allocated (assigning to more 10 users than the initial 30 users). The experimental findings demonstrate that utilitarianism and dealism do not adhere to Axiom 5 (Resource Monotonicity). Conversely, egalitarianism exhibits the ability to enhance the utility of both items in the presence of resource changes.

## 4.2 $\alpha$-RANK ALGORITHM

In this section, we will introduce $\alpha$-*rank* approach to efficiently handle the $\alpha$-fairness objective optimization in equation 2. The overall algorithm workflow can be seen in Algorithm 1.

We observe that directly optimizing the equation 2 requires huge computational costs since it is a non-linear, large-scale, and integral programming (Bertsekas, 1997). Therefore, firstly, we construct an easy-solved standard cooperative game programming (equation 3), which is the upper bound function of equation 2. Then we apply the transport optimal (OT) projection method to obtain the final ranking result (equation 4). Finally, we prove a theoretical result to show the maximum social

---

**Algorithm 1:** Algorithm of $\alpha$-*rank*

---

**Input:** User set $\mathcal{U}$, item set $\mathcal{I}$, ranking size $K$, fairness coefficient $\alpha$, OT coefficient $\lambda$, item weight $\gamma_i, \forall i \in \mathcal{I}$, user-item score $w_{u,i}, \forall u \in \mathcal{U}, \forall i \in \mathcal{I}$

**Output:** The ranking result $L_K(u), \forall u \in \mathcal{U}$

1: Get the optimal averaged exposure $\boldsymbol{e}^*$ from equation 3.
2: Initialize $\boldsymbol{m} = K\mathbf{1}$, $\boldsymbol{n} = \boldsymbol{e}^*$, $\boldsymbol{C}_{u,i} = \gamma_i w_{u,i}, \forall u \in \mathcal{U}, \forall i \in \mathcal{I}$, $\boldsymbol{B} = e^{\frac{-C}{\lambda}}$
3: **for** $t = 1, \cdots, T$ **do**
4:    $\boldsymbol{m} = K\mathbf{1} \oslash \boldsymbol{Bn}$
5:    $\boldsymbol{n} = \boldsymbol{e}^* \oslash \boldsymbol{Bm}$
6: **end for**
7: $\widetilde{\boldsymbol{x}} = \text{diag}(\boldsymbol{m})\boldsymbol{B}\text{diag}(\boldsymbol{n})$
8: $L_K(u) = \arg\max_{S \subset \{1,2,\ldots,|\mathcal{I}|\}, |S|=K} \sum_{i \in S} \widetilde{\boldsymbol{x}}_{u,i}, \quad \forall u \in \mathcal{U}$

---

utility loss across different fairness degree, named price of fairness (POF) of ranking (Bertsimas et al., 2011; 2012).

### 4.2.1 UPPER BOUND FUNCTION CONSTRUCTION

**Theorem 2** *There exists $\tau > 0$, s.t. we have the following function*

$$\hat{W}(\alpha) = \max_{\boldsymbol{e}} \sum_i \gamma_i \eta_i g(\boldsymbol{e}; \alpha)$$

$$s.t. \quad \sum_{i \in \mathcal{I}} \boldsymbol{e}_i = K, \quad 0 \leq \boldsymbol{e}_i \leq 1, \quad \eta_i = \tau \sum_{u \in \mathcal{U}} w_{u,i}, \forall i \in \mathcal{I} \tag{3}$$

$$g(\boldsymbol{e}; \alpha) = \begin{cases} \sum_i \frac{\boldsymbol{e}_i^{1-\alpha}}{1-\alpha} & \text{if } \alpha > 0, \alpha \neq 1 \\ \sum_i \log(\boldsymbol{e}_i) & \text{if } \alpha = 1 \end{cases},$$

*where $\hat{W}(\alpha) \geq \max_{\boldsymbol{v} \in \mathcal{U}} f(\boldsymbol{v}; \alpha)$ and the variable $\boldsymbol{e}_i = \frac{1}{|\mathcal{U}|} \sum_{u \in \mathcal{U}} \boldsymbol{x}_{u,i}$, which is the averaged exposure of certain item $i$ within a period of time.*

The proof of Theorem 2 can be seen in Appendix B. The optimal value $\boldsymbol{e}^*$ represents the average exposure of items achieved under the $\alpha$-fairness optimization objective. Then we will apply the Sinkhorn algorithm (Pham et al., 2020) to project the averaged exposure $\boldsymbol{e}^*$ to recommendation list $\boldsymbol{x} \in \{0,1\}^{|\mathcal{U}| \times |\mathcal{I}|}$ discussed in Section 3.

### 4.2.2 OPTIMAL TRANSPORT PROJECTION

We obtain the final ranking result by utilizing the following sample process, where $\widetilde{\boldsymbol{x}}$ (i.e. ranking score distribution) is derived from the OT projection process.

$$L_K(u) = \arg\max_{S \subset \{1,2,\ldots,|\mathcal{I}|\}, |S|=K} \sum_{i \in S} \widetilde{\boldsymbol{x}}_{u,i}, \quad \forall u \in \mathcal{U}. \tag{4}$$

We construct a matrix $\boldsymbol{C} = \mathbb{R}^{|\mathcal{U}| \times |\mathcal{I}|}$, where the element $\boldsymbol{C}_{u,i} = \gamma_i w_{u,i}$. An OT problem can be formulated as:

$$\widetilde{\boldsymbol{x}} = \arg\min_{\boldsymbol{x} \geq 0} \langle \boldsymbol{x}, -\boldsymbol{C} \rangle + \lambda H(\boldsymbol{x}) \quad \text{s.t.} \quad \boldsymbol{x}\mathbf{1} = K\mathbf{1}, \quad \mathbf{1}^\top \boldsymbol{x} = \boldsymbol{e}^*, \tag{5}$$

where $\mathbf{1}$ denotes a vector of ones, $\boldsymbol{e}^*$ denotes the optimal value of equation 3 and $\lambda$ is the coefficient of entropy regularizer. $\langle \boldsymbol{x}, -\boldsymbol{C} \rangle$ results transport plan lies on the Pareto frontier. $H(\boldsymbol{x}) = \sum_u \sum_i \boldsymbol{x}_{u,i} \log(\boldsymbol{x}_{u,i})$, which forces the variable $\boldsymbol{x}_{u,i}$ into the feasible region $[0,1]$. The constraint condition ensures that the ranking satisfies the limitation that each user can only be ranked among the top $K$ items, and it also guarantees that the exposure of each item aligns optimally with the predefined exposure vector $\boldsymbol{e}^*$.

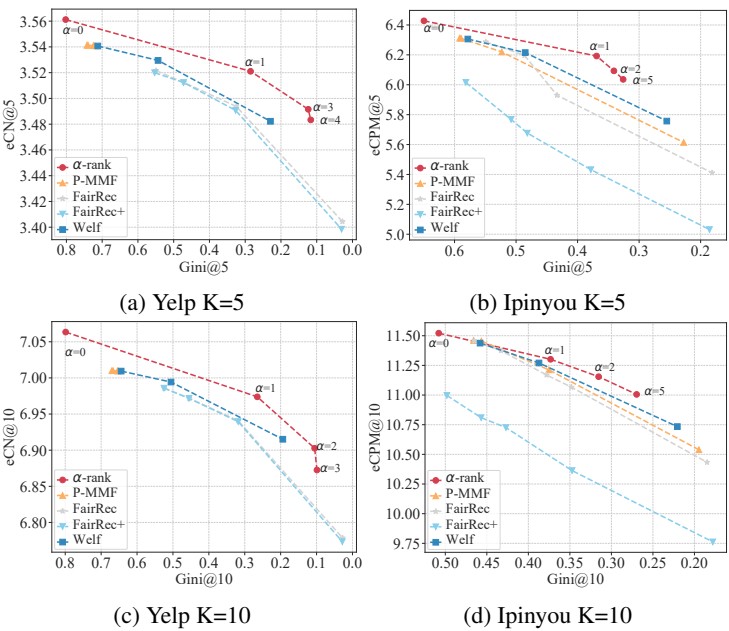

Figure 2: Pareto frontier of four different datasets with different top-$K$ ranking.

This problem can be efficiently solved by the Sinkhorn algorithm (Swanson et al., 2020), where the solution of the form $\widetilde{\boldsymbol{x}} = \text{diag}(\boldsymbol{m})\boldsymbol{B}\text{diag}(\boldsymbol{n})$, where $\text{diag}(\cdot)$ denote the generating diagonal matrix from vector, $\boldsymbol{B} = e^{\frac{-C}{\lambda}}$, and $\boldsymbol{m} \in \mathbb{R}^{|\mathcal{U}|}$, $\boldsymbol{n} \in \mathbb{R}^{|\mathcal{I}|}$, which iteratively computes

$$\boldsymbol{m} \leftarrow K\mathbf{1} \oslash \boldsymbol{B}\boldsymbol{n}, \quad \boldsymbol{n} \leftarrow \boldsymbol{e}^* \oslash \boldsymbol{B}\boldsymbol{m},$$

where $\oslash$ denotes element-wise division.

### 4.2.3 Price of Item Fairness

Typically, when conducting fairness adjustments in ranking, it may result in the redistribution of resources that can lead to a reduction in the total utilities ($\sum_i \boldsymbol{v}_i$) of the system. In this section, we aim to bound the price of item fairness (POF) (Bertsimas et al., 2011), which measures the maximum social utility loss across different fairness degrees, i.e. different $\alpha$ values.

**Theorem 3** *The price of item fairness is quantified as the relative reduction in the sum of utilities when comparing the fair solution to the utilitarian solution, represented as:*

$$POF = \frac{W(0) - W(\alpha)}{W(0)} \leq 1 - O(|\mathcal{U}|^{-\frac{\alpha}{1+\alpha}}), \tag{6}$$

**Remark 3** *The Theorem 3 holds that when increasing the item fairness degrees ($\alpha$ becomes larger) in a ranking system, there is a bound on the rate $1 - O(|\mathcal{U}|^{-\frac{\alpha}{1+\alpha}})$ at which utilities will decrease.*

## 5 Experiment

We evaluate the performance of $\alpha$-*rank*. In the experiment, we mainly conduct the CTR/CVR-based fairness discussed in Table 1. For the exposure-based fairness, please see the Appendix I. The source code and experiments have been shared in supplementary file.

### 5.1 Experimental settings

**Dataset.** The experiments were based on two large-scale, publicly available ranking applications, including: **Yelp**[1]: a large-scale businesses recommendation dataset. It has 154543 samples, which

---

[1]https://www.yelp.com/dataset

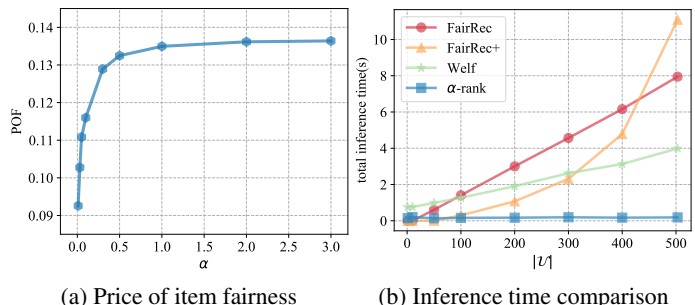

(a) Price of item fairness        (b) Inference time comparison

Figure 3: Sub-figure (a) illustrates the price of item fairness (POF) change w.r.t fairness degree $\alpha$. Sub-figure (b) describes online inference items for $\alpha$-*rank* and other baselines w.r.t user size $|\mathcal{U}|$.

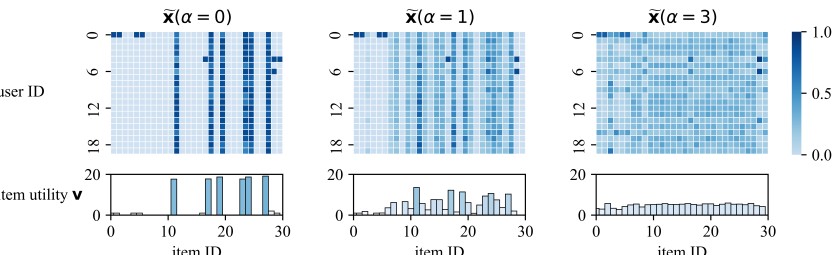

Figure 4: Visualization of $\alpha$-*rank* result.

contains 17034 users, 11821 items. **Ipinyou** (Liao et al., 2014)[2]: a large-scale advertising dataset. We only used the clicked data, which contains 18588 samples, which contains 18565 users, 149 advertisements. Every advertisement has a bidding price.

During the pre-processing step, users and items that had interactions with fewer than 5 items or users were excluded from the entire dataset to mitigate the issue of extreme sparsity. Following (Zhang et al., 2022; Xu et al., 2023b), we used BPR (Rendle et al., 2012) model to compute the CTR-CVR value of user-item pair. For each user-item pair $(u, i)$, the model will output the CTR-CVR value $w_{u,i}$. For the item weight $\gamma_i$, a value of 1 is assigned for recommendation applications, while for advertising applications, $\gamma_i = \log(\text{bid}_i)$, where $\text{bid}_i$ represents the bidding price of an advertisement.

**Evaluation.** As for the evaluation metrics, the performances of the models were evaluated from two aspects: social welfare, and fairness degree. As for the social welfare, following the practices in (Wu et al., 2021; Xu et al., 2023b; Yang et al., 2019), we utilized excepted Click/Conversion Number (eCN) for recommendation application and expected Cost Per Mile (eCPM) for advertising application under top-$K$ ranking.:

$$\text{eCN@K} = \frac{1}{|\mathcal{U}|} \sum_{i \in \mathcal{I}} \boldsymbol{v}_i, \quad \text{eCPM@K} = \frac{1}{|\mathcal{U}|} \sum_{i \in \mathcal{I}} \text{bid}_i \boldsymbol{v}_i. \tag{7}$$

.

As for the fairness degree, we utilized the Gini Index (Do & Usunier, 2022; Do et al., 2021), which is the most common measure of item utility inequality under top-$K$ ranking. Formally, it defines as:

$$\text{Gini@K} = \frac{\sum_i \sum_j |\gamma_i \boldsymbol{v}_i - \gamma_j \boldsymbol{v}_j|}{2|\mathcal{I}| \sum_i \gamma_i \boldsymbol{v}_i}, \tag{8}$$

where it ranges from 0 to 1, with 0 representing perfect equality (every item has the same utility), and 1 representing perfect inequality (one item has all the utility, while every item else has none).

**Baselines.** The following representative item fairness models were chosen as the baselines:

**FairRec** (Patro et al., 2020) and **FairRec+** (Biswas et al., 2021) proposed to ensure Max-Min Share ($\alpha$-MMS) of exposure for the items. **Welf** (Do et al., 2021) use the Frank-Wolfe algorithm to maximize the Welfare functions of worst-off items. **P-MMF** (Xu et al., 2023a) utilized the mirror descent method to improve the worst-off item's utility.

---

[2]http://contest.ipinyou.com/

## 5.2 EXPERIMENT RESULTS

Figure 2 shows the Pareto frontiers Xu et al. (2023a) of Gini Index (abbreviated as Gini.) and eCN/eCPM on two application datasets with different ranking size $K$. The Pareto frontiers were constructed by systematically adjusting various parameters of the models and then selecting the points with the best performance in terms of both Gini@K and eCN@K/eCPM@K, resulting in an optimized trade-off between item fairness and total utilities.

Analyzing the Pareto frontiers, it becomes evident that the proposed $\alpha$-*rank* method consistently outperforms the baseline methods (as indicated by the $\alpha$-*rank* curves occupying the upper right corner). This Pareto dominance signifies that, for a given eCN@K/eCPM@K level, $\alpha$-*rank* achieves superior Gini@K values, and for a given Gini@K level, it attains better eCN@K/eCPM@K performance. These results highlight that $\alpha$-*rank* significantly outperforms the baseline methods.

## 5.3 EXPERIMENT ANALYSIS

We also conducted experiments to analyze $\alpha$-*rank* on Yelp for Top-10 ranking. For ablation studies and Lorenz curve Gastwirth (1971) analysis, please see Appendix H and Appendix G, respectively.

**Price of item fairness.** Firstly, we conducted an experiment to demonstrate how the price fairness of the item in Figure 3 (a) changes with respect to variations in the fairness degree $\alpha$, ranging from 0.0 to 3.0. we directly compute the POF based on equation 6. From the curve, it is evident that as we increase the fairness degree $\alpha$, the $\alpha$-*rank* approach leads to a reduction in the total utilities of items. The experiment verified the theoretical analysis results in Theorem 3.

**Inference time.** We conducted experiments to investigate the total inference time of the $\alpha$-*rank* method compared to other item fairness baselines. In our analysis, our objective is to assess the total inference time across various user sizes $|\mathcal{U}|$, within real-world ranking applications. Therefore, we conducted tests to measure the total inference time of various models in relation to the varying number of users, all while keeping the number of items constant.

Figure 3 (b) reports the curves of total inference time (s) w.r.t. user size $|\mathcal{U}|$. It's worth noting that the $\alpha$-*rank* method exhibits a remarkably low inference time, typically taking less than ten million seconds across different user sizes. Furthermore, when compared to other baseline methods, the inference time of these alternatives tends to increase either linearly or exponentially with changing user sizes, whereas $\alpha$-*rank* consistently maintains a low inference time. The $\alpha$-*rank* method involves matrix operations with limited sensitivity to changes in user size.

**Visualizing ranking results.** In Figure 4, we visualize the ranking result matrix $\widetilde{\boldsymbol{x}}$ and the utility vector $\boldsymbol{v}$ of items generated by the $\alpha$-*rank* method for different values of $\alpha$ (0, 1, and 3), where these values correspond to utilitarianism, dealism, and egalitarianism, respectively. The distribution of vector $\boldsymbol{v}$ reflects the fairness degree of items. The detailed histogram of utility level of items under different $\alpha$ can be seen in Appendix F.

The results clearly demonstrate that the utilitarianism solution consistently ranks the most popular items highly for users, thereby enhancing overall utility but potentially leading to market dominance by a few top items. Regarding dealism, $\alpha$-*rank* approach tends to distribute rankings to items in proportion to their contribution to the market. For egalitarianism, $\alpha$-*rank* method strives to provide equal exposure and similar utilities to every item in the ranking. The experiment also served as validation that $\alpha$-*rank* method can effectively adapt to various fairness principles as intended.

## 6 CONCLUSION

This paper proposes $\alpha$-*rank* model that aims to unify the item fairness in ranking from the cooperative game theory view. Firstly, we conducted an analysis of various fairness principles in ranking and unified these principles within the framework of cooperative game theory. Then we introduced the approach of $\alpha$-*rank* can well-balance different fairness principles. Theoretical results to establish the maximum total utility loss for different values of $\alpha$. Finally, Experiment results show that $\alpha$-*rank* can outperform the state-of-the-art baselines efficiently and effectively.

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

# A PROOF OF THEOREM 1

## A.1 AXIOM 1

Suppose exist another solution $v \in \mathcal{U}$ so that utility vector $v \geq v^f$ and $v^f \neq v$.

For the utilitarianism, we have
$$\mathbf{1}^\top v \geq \mathbf{1}^\top v^f, \quad v^f \neq v,$$
which verifies that $v$ is a better solution compared to $v^f$. There is a contradiction. Therefore, utilitarianism satisfies the Axiom 1.

For the dealism, we have
$$\mathbf{1}^\top \log(v) \geq \mathbf{1}^\top \log(v^f), \quad v^f \neq v,$$
which also verifies that $v$ is a better solution compared to $v^f$. There is a contradiction. Therefore, dealism satisfies the Axiom 1.

For the egalitarianism, we have
$$v_i = v_j, \forall i, j \in \mathcal{I}, v \in \mathcal{U}.$$
Therefore, if $v \geq v^f$ and $v^f \neq v$, there must be
$$v_i > v_i^f, \forall i \in \mathcal{I},$$
and therefore,
$$\min_i v_i > \min_i v_i^f,$$
which is a contradiction with the optimization objective. Therefore, egalitarianism satisfy the Axiom 1.

Q.E.D.

## A.2 AXIOM 2

As the formulation doesn't take item attributes into account and applies the same operation to all items, it becomes evident that the Axiom holds.

## A.3 AXIOM 3

For utilitarianism, consider the following CTR matrix for recommendation,
$$\begin{bmatrix} 0.1 & 0.2 & 0.3 \\ 0.5 & 0.4 & 0.1 \\ 0.7 & 0.5 & 0.3 \end{bmatrix},$$
where the element in i-th row and j-column is $w_{u,i}$ and ranking size is 1. In such a matrix, the utilitarianism solution is $0.3 + 0.5 + 0.7 = 1.5$ since it always chooses the highest value among the items of certain user (i.e. selects the highest value of one row). However, we apply the transformation $A$, such that the matrix becomes
$$\begin{bmatrix} 0.1*4 & 0.2 & 0.3 \\ 0.5*4 & 0.4 & 0.1 \\ 0.7*4 & 0.5 & 0.3 \end{bmatrix},$$
then the utilitarianism solution is $0.4 + 2 + 2.8 \neq 4 * 1.5$. Therefore, utilitarianism does not satisfy Axiom 3.

For dealism, for the transformation $A$, we have
$$f(A(v)) = \max \sum_i \log(c_i v_i) = \max \sum_i \log(v_i) = f(v).$$

Therefore, dealism satisfies Axiom 3.

For egalitarianism, it is easy to observe if could happen
$$\arg\min_i v_i \neq \arg\min_i c_i v_i,$$
therefore, egalitarianism does not satisfy Axiom 3.

### A.4 AXIOM 4

It is evident that all three optimization objectives do not take resource available levels into consideration. In simpler terms, if the resources allocated to an item are reduced, and the initial solution still falls within the feasible region, the solution remains unchanged from the original.

### A.5 AXIOM 5

For the utilitarianism, consider the matrix

$$\begin{bmatrix} 0.1 & 0.2 & 0.3 \\ 0.5 & 0.4 & 0.1 \end{bmatrix},$$

where the element in i-th row and j-column is $w_{u,i}$ and ranking size is $1$. In the context of ranking applications, we assume that each item is exposed at most once. Therefore, the ranking model will choose $0.3$ in the first row and $0.5$ in the second row. However, if we increase the resource to

$$\begin{bmatrix} 0.1 & 0.2 & 0.3 \\ 0.5 & 0.3 & 0.1 \\ 0.7 & 0.1 & 0.1 \end{bmatrix},$$

the ranking model will choose $0.3$ in the first row, $0.3$ in the second row and $0.7$ in the third row. The first item's utility decreases from $0.4$ to $0.3$. Therefore, utilitarianism does not satisfy Axiom 5.

Similarly, dealism does not satisfy Axiom 5.

For egalitarianism, $\mathcal{D}_1 \subset \mathcal{D}_2$ and $\mathcal{D}_1 \neq \mathcal{D}_2$, we have

$$\boldsymbol{v}_i = \boldsymbol{v}_j, \forall i, j \in \mathcal{I}, \boldsymbol{v} \in \mathcal{U},$$

and since it satisfies the Axiom 1, we therefore have

$$\arg\max_{\boldsymbol{v} \in \mathcal{D}_1} f(\boldsymbol{v}) \leq \arg\max_{\boldsymbol{v} \in \mathcal{D}_2} f(\boldsymbol{v}).$$

Q.E.D.

## B PROOF OF THEOREM 2

According to the equation 1, we have

$$W(\alpha) = \max f(\boldsymbol{v}; \alpha),$$
$$\text{s.t.} \quad \mathbf{1}^\top \boldsymbol{x}_u = K, \quad \forall u \in \mathcal{U}$$
$$\boldsymbol{x}_{u,i} = \{0, 1\}, \quad \forall i \in \mathcal{I}, u \in \mathcal{U}$$
$$\boldsymbol{v}_i = \gamma_i \sum_{u \in \mathcal{U}} w_{u,i} \boldsymbol{x}_{u,i}, \quad \forall i \in \mathcal{I}$$

Let abbreviate $|\mathcal{U}| = N$ and $\boldsymbol{e}_i = \frac{1}{N} \sum_{u \in \mathcal{U}} \boldsymbol{x}_{u,i}$.

Therefore, we can relax the first condition as

$$\sum_{u \in \mathcal{U}} \sum_{i \in \mathcal{I}} \boldsymbol{x}_{u,i} = NK,$$

i.e.

$$\sum_{i \in \mathcal{I}} e_i = K \tag{9}$$

We also relax the second condition as

$$0 \leq \sum_{u \in \mathcal{U}} \boldsymbol{x}_{u,i} \leq N, \forall i \in \mathcal{I}$$

i.e.

$$0 \leq \boldsymbol{e}_i \leq 1, \forall i \in \mathcal{I}. \tag{10}$$

Finally, their exists $\tau > 0,$, we can relax the third condition as

$$\sum_{u \in \mathcal{U}} w_{u,i} \boldsymbol{x}_{u,i} \leq \tau \sum_{u \in \mathcal{U}} w_{u,i} \sum_{u \in \mathcal{U}} \boldsymbol{x}_{u,i},$$

i.e.

$$\sum_{u \in \mathcal{U}} w_{u,i} \boldsymbol{x}_{u,i} \leq N \eta_i \boldsymbol{e}_i, \tag{11}$$

where $\eta_i = \tau \sum_{u \in \mathcal{U}} w_{u,i}$.

Combining Equation 9, 10, 11, we have $\hat{W}(\alpha) \geq W(\alpha)$.

Q.E.D

## C  LEMMA 1

**Lemma 1** *Given the vector $\boldsymbol{x} \in \mathbb{R}^N$, $\sum_{i=1}^{N} w_i \boldsymbol{x}_i^{1-\alpha} \geq C$, for any $\alpha > 0, w_i > 0, \boldsymbol{x}_i > 0$, we have*

$$\sum_{i=1}^{N} w_i \boldsymbol{x}_i \geq C \sigma^{\alpha} N^{-\frac{\alpha}{1+\alpha}}, \tag{12}$$

*where $\sigma = \min_i \boldsymbol{x_i}$.*

### C.1  PROOF OF LEMMA 1

We can observe that $f(x) = x^{1+\alpha}$ is a concave function, therefore, we apply the Hölder inequality, we have when $\frac{1}{1+\alpha} + \frac{\alpha}{1+\alpha} = 1$,

$$C \leq \sum_{i=1}^{N} w_i \boldsymbol{x}_i * \frac{1}{\boldsymbol{x}_i^{\alpha}} \leq (\sum_{i=1}^{N} w_i^{1+\alpha} \boldsymbol{x}_i^{1+\alpha})^{\frac{1}{1+\alpha}} (\sum_{i=1}^{N} \frac{1}{\boldsymbol{x}_i^{1+\alpha}})^{\frac{\alpha}{1+\alpha}} \tag{13}$$

From concave attribute, we have

$$(\sum_{i}^{N} w_i \boldsymbol{x}_i)^{1+\alpha} \geq \sum_{i=1}^{N} w_i^{1+\alpha} \boldsymbol{x}_i^{1+\alpha},$$

therefore, we have

$$(\sum_{i=1}^{N} w_i^{1+\alpha} \boldsymbol{x}_i^{1+\alpha})^{\frac{1}{1+\alpha}} \leq \sum_{i=1}^{N} w_i \boldsymbol{x}_i \tag{14}$$

Let $\sigma = \min_i \boldsymbol{x}_i$, we have

$$(\sum_{i=1}^{N} \frac{1}{\boldsymbol{x}_i^{1+\alpha}})^{\frac{\alpha}{1+\alpha}} \leq \frac{N^{\frac{\alpha}{1+\alpha}}}{\sigma^{\alpha}} \tag{15}$$

Then combining equation 14 and equation 15 the equation 13 becomes:

$$\sum_{i=1}^{N} w_i x_i \geq C \sigma^{\alpha} N^{-\frac{\alpha}{1+\alpha}}$$

Q.E.D

## D  PROOF OF THEOREM 3

To simply the symbol, let $m_i = \gamma_i \eta_i$. The objective becomes:

$$\hat{W}(\alpha) = \max_{\boldsymbol{e}} \sum_i m_i \frac{\boldsymbol{e}_i^{1-\alpha}}{1-\alpha}$$
$$\text{s.t.} \quad \sum_i \boldsymbol{e}_i = K \tag{16}$$
$$0 \le \boldsymbol{e}_i \le 1$$

, where the input $\boldsymbol{e}$ optimal value is $\boldsymbol{z}$, which represents be the best allocation under the $\alpha$-fairness criterion.

Firstly, we will bound the $\hat{W}^\alpha$: without generality, we assume that:

$$m_1 \boldsymbol{z}_1 \ge m_2 \boldsymbol{z}_2 \ge \cdots, \ge m_{|\mathcal{I}|} \boldsymbol{z}_{|\mathcal{I}|}. \tag{17}$$

The necessary first-order condition for the optimality of $\boldsymbol{e}$ can be expressed as:

$$\nabla \hat{W}^\alpha(\boldsymbol{z})(\boldsymbol{e} - \boldsymbol{z}) \le 0, \forall \boldsymbol{e} \in \mathcal{E},$$

where

$$\mathcal{E} = \{\boldsymbol{e} | \sum_i \boldsymbol{e}_i = K, \quad 0 \le \boldsymbol{e}_i \le 1\}.$$

The equation can be equivalently written as:

$$\boldsymbol{g}^\top \boldsymbol{e} \le 1, \forall \boldsymbol{e} \in \mathcal{E}, \tag{18}$$

where

$$\boldsymbol{g}_i = \frac{m_i \boldsymbol{z}_i^{-\alpha}}{\sum_i m_i \boldsymbol{z}_i^{1-\alpha}}. \tag{19}$$

We observe the Equation equation 18, which is a well-studied knapsack problem (Salkin & De Kluyver, 1975), with the best solution:

$$\frac{\sum_{k=1}^K m_{|\mathcal{I}|-k+1} \boldsymbol{z}_{|\mathcal{I}|-k+1}^{-\alpha}}{\sum_i m_i \boldsymbol{z}_i^{1-\alpha}} \le 1, \tag{20}$$

since according the equation 17, we have

$$m_1 \boldsymbol{z}_1^{-\alpha} \le m_2 \boldsymbol{z}_2^{-\alpha} \le \cdots, \le m_{|\mathcal{I}|} \boldsymbol{z}_{|\mathcal{I}|}^{-\alpha}.$$

From the equation 20, exists $0 < \lambda < 1$, we have:

$$\sum_i m_i \boldsymbol{z}_i^{1-\alpha} \ge \lambda W^0 \boldsymbol{z}_1^{-\alpha}, \tag{21}$$

since

$$\sum_{k=1}^K m_{|\mathcal{I}|-k+1} \boldsymbol{z}_{|\mathcal{I}|-k+1}^{-\alpha} \ge \boldsymbol{z}_1^{-\alpha} \lambda \sum_{k=1}^K m_{[k]} = \lambda W^0 \boldsymbol{z}_1^{-\alpha},$$

where $m_{[k]}$ denotes the $k$-th largest element of $m_i$.

Taking the equation 21 into Lemma 1, we have

$$\sum_i^{|\mathcal{I}|} m_i \boldsymbol{z}_i \ge \lambda W^0 \left(\frac{\boldsymbol{z}_{|\mathcal{I}|}}{\boldsymbol{z}_1}\right)^\alpha N^{-\frac{\alpha}{1+\alpha}}. \tag{22}$$

Therefore,

$$\text{POF} = \frac{W(0) - W(\alpha)}{W(0)} \le \frac{W(0) - \hat{W}(\alpha)}{W(0)}$$
$$\le 1 - \lambda \left(\frac{\boldsymbol{z}_{|\mathcal{I}|}}{\boldsymbol{z}_1}\right)^\alpha N^{-\frac{\alpha}{1+\alpha}}$$
$$= 1 - O(N^{-\frac{\alpha}{1+\alpha}}).$$

Q.E.D.

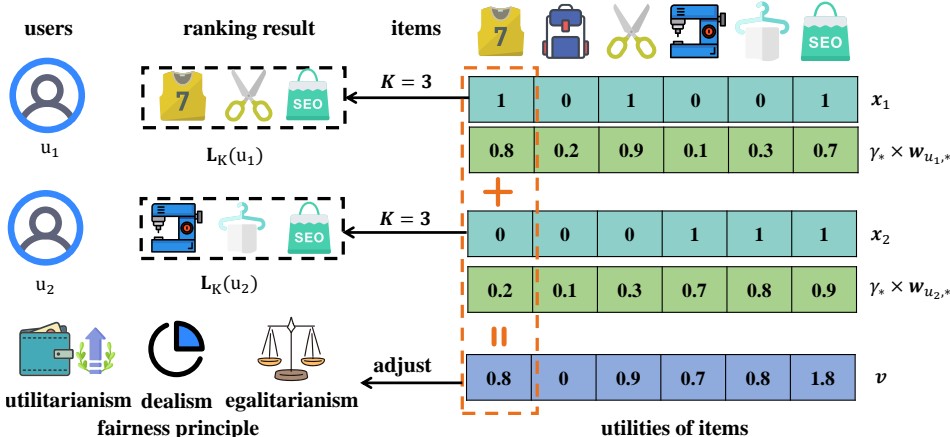

Figure 5: An illustrative example to visualize item fairness in ranking applications.

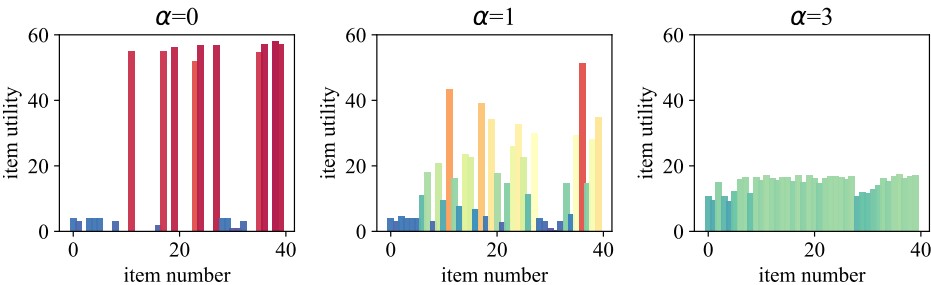

Figure 6: Visualization of items utilities for different $\alpha$.

# E  AN ILLUSTRATIVE EXAMPLE OF ITEM FAIRNESS

In this section, we will give an illustrative example of item fairness.

Figure 5 provides an illustrative example of item fairness in the ranking process. In this scenario, we assume the recommender system includes users $u_1$ and $u_2$, and there are six items in the RS available for recommendation to these users.

In a simple way, we suppose $u_1$ and $u_2$, and the system ranks a list of $K = 3$ items from six item candidates. The ranking model will make the ranking list $L_K(u_1), L_K(u_2)$ based on the decision variable $x$. For example, in the case of user $u_1$, the recommendation will expose items with a value of 1 for the decision vector $x$.

We also set the weight $\gamma_i \times w_{u,i}$ of item $i$. The utilities $v$ of items are determined by aggregating the scores of the exposed items across the entire user set. For example, for the first item $i_1$ (yellow cloth), the utility $v_1$ is computed as $1 * 0.8 + 0 * 0.2 = 0.8$. We aim to balance the utilities of items from utilitarianism, dealism, and egalitarianism, respectively.

# F  VISUALIZATION OF ITEM UTILITY

To give a better understanding of how the hyperparameter $\alpha$ influences fairness under different ethical principles (utilitarianism, dealism, and egalitarianism), we plotted histograms of item utility for each of the different values of $\alpha$ (0, 1, and 3).

As illustrated in Figure 6, when $\alpha$ increases from 0 to 3, the maximum utility of items decreases, while the minimum utility increases. This indicates a shift toward a more equitable and fair distribution of utility among items.

When $\alpha = 0$ (corresponding to utilitarianism), the difference in utility level between items is very large(more than 56.9). When $\alpha = 3$ (corresponding to egalitarianism), utility levels converge, and the difference between the highest and lowest utility narrows to a mere 8.34.

Obviously, when $\alpha$ is small, $\alpha$-*rank* focuses more on maximizing the overall utility, allocating most exposure toward the item with the largest $w_u i$. On the contrary, for larger values of $\alpha$, $\alpha$-*rank* places greater emphasis on fairness among items. When $\alpha$ falls in between, $\alpha$-*rank* strives to achieve a balance between accuracy and fairness, ensuring that more items receive adequate exposure while enhancing the overall utility of the item group.

## G  LORENZ CURVE OF DIFFERENT FAIRNESS MODELS

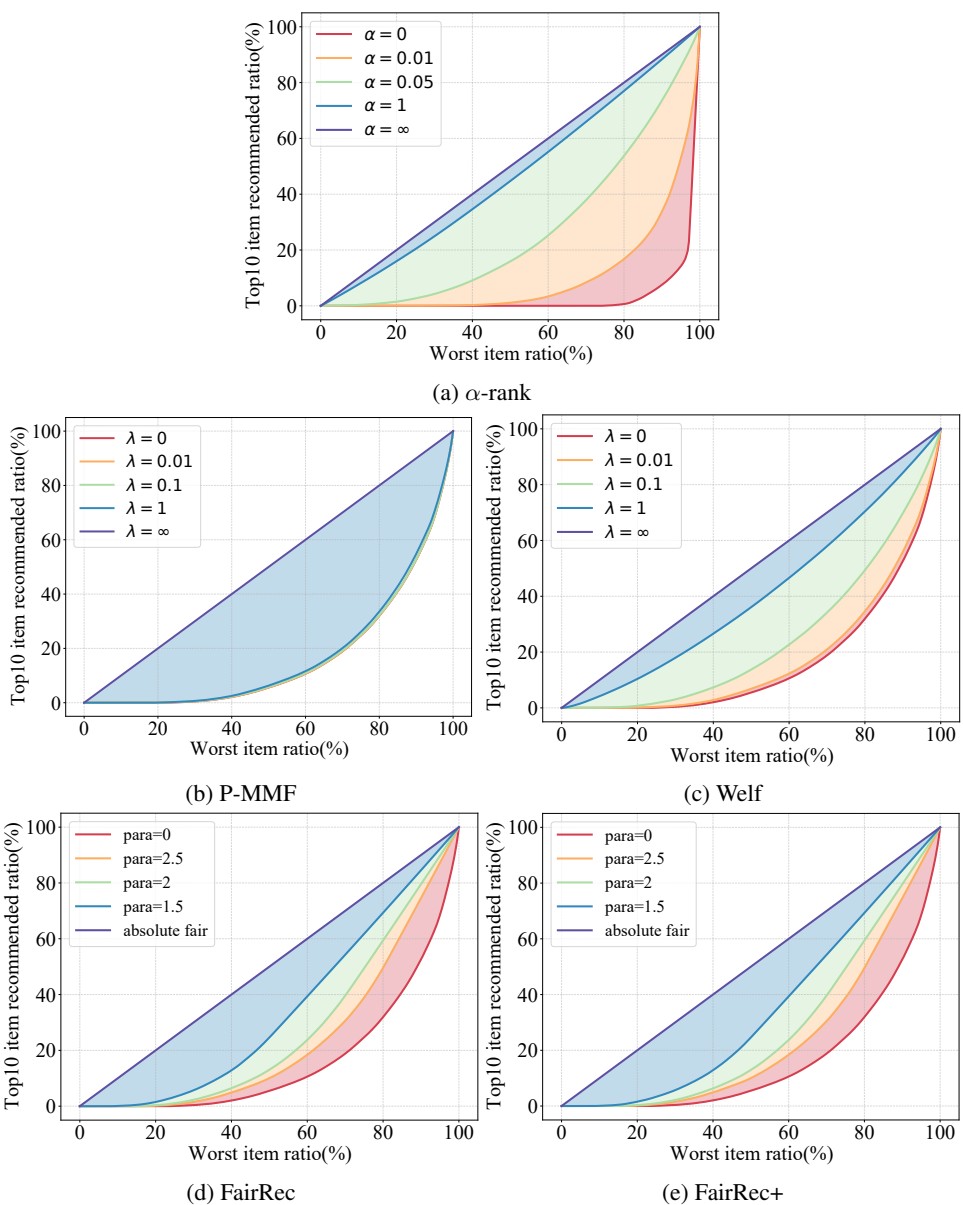

Figure 7: The Lorenz curves for different models on item exposure.

To comprehensively analyze the item fairness of $\alpha$-*rank* and other baselines, we conducted an analysis employing Lorenz curves Gastwirth (1971), as shown in Figure 7, to visualize the distribution of item utility. Lorenz curve is a well-established tool for assessing distributional inequality in dif-

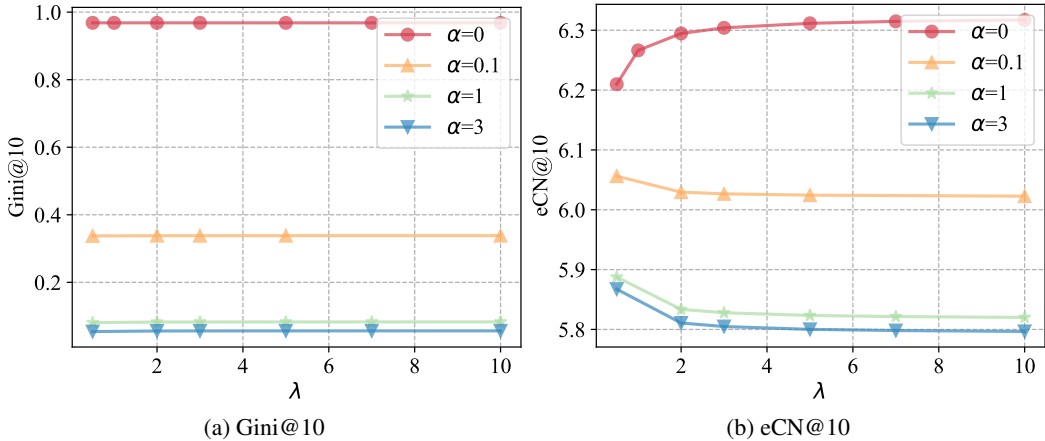

Figure 8: Gini and eCN change w.r.t the coefficient of entropy regularizer $\lambda$ in OT under different $\alpha$

ferent scenarios. The gini coefficient is the area between the Lorenz curve and the absolute fairness line.

Here are the steps we took to plot the Lorenz curve. Firstly, we sort the items according to their utility level, from low to high. Then, we calculate the percentage of total utility for each percentage of items separately. Finally, we plot these percentage data as a curve that shows the proportion of utility (y%) occupied by the worst x% of items. Note that the x-axis represents the cumulative percentage of item numbers, while the y-axis represents the cumulative utility percentage. It is crucial to highlight that the diagonal line extending towards the upper right corner signifies a state of absolute fairness.

We can see that in Figure 7 (a), when $\alpha \rightarrow 0$, $\alpha$-rank predominantly emphasizes total utility level, and the fairness shows poor performance. In contrast, as $\alpha \rightarrow \infty$, $\alpha$-rank increasingly prioritizes fairness, resulting in a distribution of item utility that tends towards egalitarianism.

We can show that influence can be exerted on the equity of utility distribution through parameter tuning of other baselines P-MMF, Welf, FairRec, and FairRec+. Compared with other baseline models, $\alpha$-rank has a greater range of control over the distribution through hyperparameter adjustments. As the parameter $\alpha$ varies from 0 to infinity, it causes the Lorenz curve to shift in a manner where all item utilities can be taken into consideration. However, other baseline methods are sensitive to their parameters, which can make them challenging to adjust with respect to the requirements of practical applications.

# H ABLATION EXPERIMENT OF $\alpha-$RANK

To better investigate the performance of our model under different parameters settings, we conducted a series of ablation experiments on the Yelp dataset under ranking size $K = 10$. Similar experiment results are also obersved on other dataset and other ranking size $K$.

## H.1 ABLATION STUDY ON OT COEFFICIENT $\lambda$

In this section, we conducted experiments using two metrics, Gini@10 and eCN@10, to investigate how the OT regularizer coefficient $\lambda$ influences model performance. We adjusted $\lambda$ value within the range of $[0, 10]$.

Figure 8 (a) illustrates how the Gini metric changes with $\lambda$. We can observe that Gini remains relatively stable across the $\lambda$ values ranging from 0 to 10. Therefore, we can conclude that $\lambda$ value is not sensitive to the fairness metric Gini. Moreover, as the parameter $\alpha$ increases, $\alpha$-rank begins to prioritize fairness among items more prominently, resulting in a decrease in Gini.

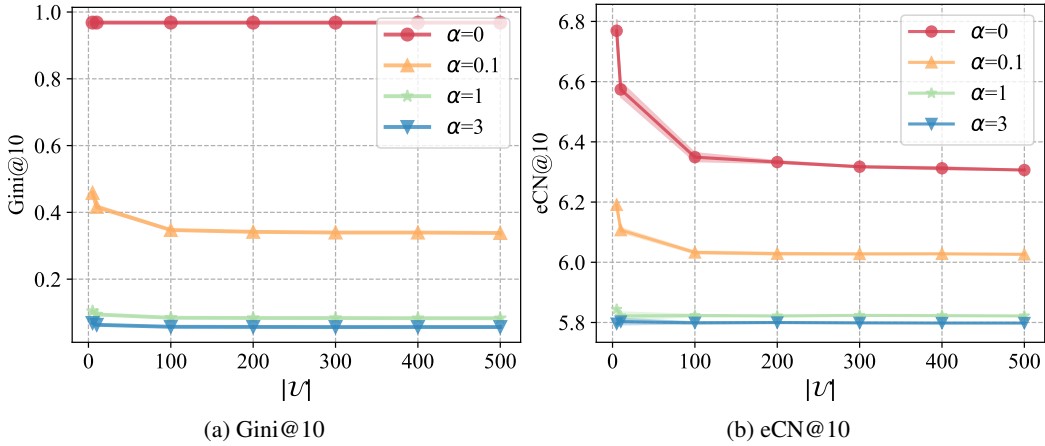

(a) Gini@10                                    (b) eCN@10

Figure 9: Gini and eCN change w.r.t user size $|\mathcal{U}|$

Curves in Figure 8 (b) report the change in eCN with respect to $\lambda$, ranging from 0 to 10. We can see that the eCN value shows a drastic change when $\lambda$ transitions from 0 to 4. Specifically, when $\alpha = 0$, eCN increases by 1.6% (from 6.21 to 6.31), and when $\alpha = 3$, eCN decreases by 1.3% (from 5.87 to 5.80). As $\alpha$ increases, the decreasing trend of eCN@10 becomes more pronounced.

In the OT problem, $\lambda$ controls the size of the entropy regularizer. By tuning the parameter $\lambda$, we can trade-off between smoothness and convexity of the exposure probability distribution among items. A higher $\lambda$ will emphasize more on the smoothness of the probability distribution, implying a reduced disparity in exposure among items. When $\alpha$ is relatively small ($\alpha \leq 0.01$), a slight reduction in the difference in exposure probability among items allows some items with lower $w_{u,i}$ to gain more exposure, resulting in an increase in eCN. However, when $\alpha$ becomes larger, the increased smoothness of probability distributions leads to a decrease in exposure for items with higher $w_{u,i}$, consequently resulting in a decrease in eCN@10.

### H.2 Ablation study on User Size $|\mathcal{U}|$

In this section, we conducted experiments on the performance of $\alpha$-*rank* with respect to user size $|\mathcal{U}|$, which is significant for understanding how $\alpha$-rank adapts to different user sizes. For the experimental setting, we randomly sampled $|\mathcal{U}| \in [5, 500]$ numbers of users. Finally, we plotted the curve based on the mean and variance of Gini@10 and eCN@10.

Figure 9 (a) illustrates that Gini@10 remains nearly constant regardless of changes in user size. This shows that $\alpha$-rank demonstrates consistent fairness performance even when confronted with varying user sizes.

However, Figure 9 (b) reveals a decline in eCN@10 as user size increases. Furthermore, the extent of eCN@10 reduction diminishes with higher $\alpha$ values. This is because when the number of users is relatively small, their interests tend to be more focused, making it easier for $\alpha$-rank to allocate exposure. As $\alpha$ increases from 0 to 3, $\alpha$-rank transitions from prioritizing accuracy to equalizing exposure across items.

### H.3 Ablation study on Item Size $|\mathcal{I}|$

Similarly, we study the influence of item size $|\mathcal{I}|$ on the performance of $\alpha$-*rank*. To conduct our analysis, we randomly sampled $|\mathcal{I}| \in [20, 300]$ numbers of items for three times and plotted the figure using the mean and variance of Gini@10 and eCN@10.

The curves, drawn in Figure 10 (a), reveal a notable trend: Gini@10 increases as item size grows. The trend is particularly pronounced when $\alpha$ is relatively small ($\alpha < 0.01$). In contrast, when $\alpha$ is relatively large (i.e. $\alpha \geq 1$), we can barely see the trend of increase, and the curves nearly overlap, where Gini@10 is extremely close to 0. This observation highlights the role of the parameter $\alpha$ in

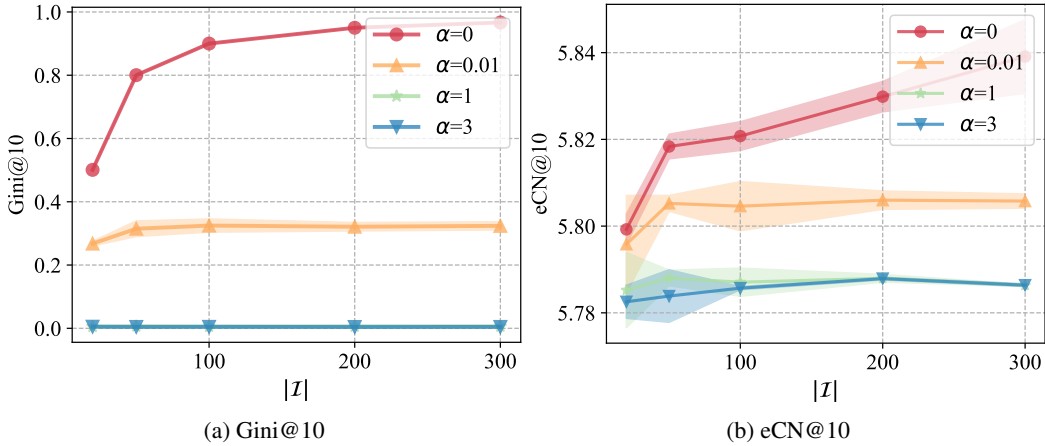

(a) Gini@10                    (b) eCN@10

Figure 10: Gini and eCN change w.r.t the item size $|\mathcal{I}|$

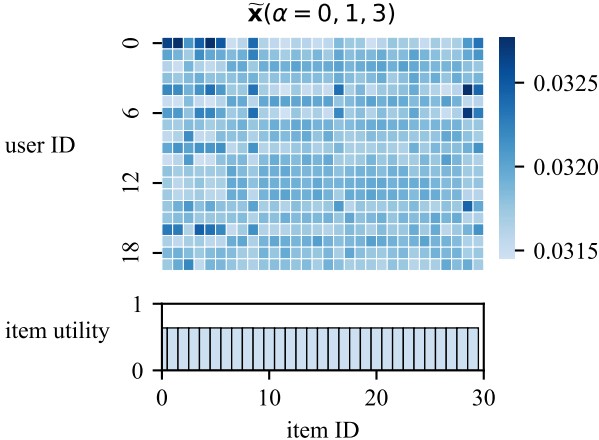

Figure 11: Visualization of exposure-based $\alpha$-rank result

stabilizing item fairness, even when dealing with large item sizes. Moreover, it demonstrates that by adjusting $\alpha$, we can maintain item fairness even if the item size is high.

In Figure 10 (b), we observed a similar pattern to Figure 10 (a). With item size $|\mathcal{I}|$ increases, eCN@10 also rises and such increase becomes more and more apparent as $\alpha$ increases. This suggests that $\alpha$-rank can achieve a more accurate and user-satisfying result when applied to larger item datasets.

## I  EXPOSURE-BASED FAIRNESS

In this section, we aim to analyze the exposure-based fairness Xu et al. (2023a); Patro et al. (2020) performance of $\alpha$-fairness and other baselines. In exposure-based fairness, we set $w_{u,i} = 1$ discussed in Table 1. The exposure-based fairness focuses on the exposure of each item as their utilities.

### I.1  VISUALIZING RANKING RESULT

In Figure 11, we visualized the ranking result matrix $\widetilde{x}$ and the utility vector $v$ of exposure-based $\alpha$-rank at different values of alpha (0, 1, and 3). As we can see in Figure 11, no matter what value of $\alpha$ we choose, exposure-based $\alpha$-rank tends to equalize item exposure and allocate utility more fairly among items.

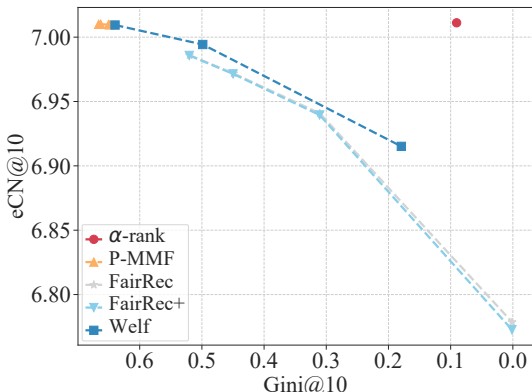

Figure 12: Pareto Frontier of exposure-based $\alpha$-rank and other baselines

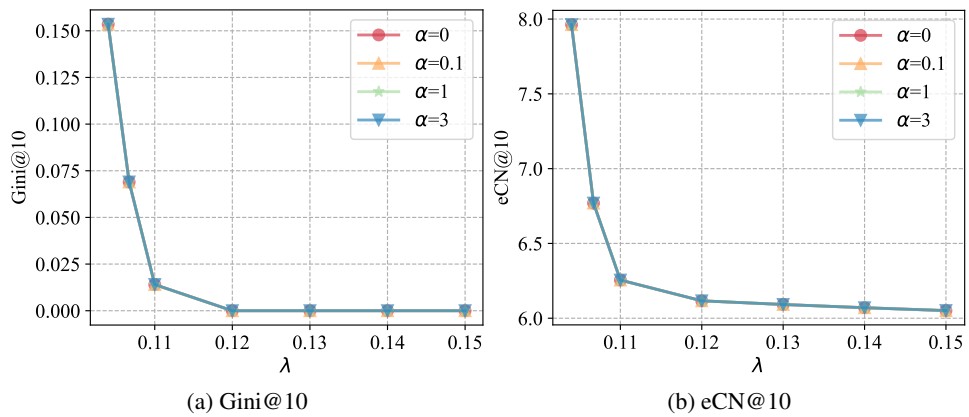

(a) Gini@10                          (b) eCN@10

Figure 13: Gini and eCN change w.r.t the coefficient of OT regularizer $\lambda$ in OT under different $\alpha$

## I.2   PARETO FRONTIER

To verify the effectiveness of $\alpha$-*rank* in exposure-based fairness, we conducted a comparative analysis by plotting the Pareto frontiers of $\alpha$-*rank* against all baseline methods. Figure 12 illustrates the performance of $\alpha$-*rank* in comparison to other baseline algorithms. From the figure, we can observe that $\alpha$-*rank* also outperforms the baselines in terms of both Gini@10 and eCN@10 metrics in the realm of exposure-based fairness. These results also verify the effectiveness of $\alpha$-rank over the baselines.

## I.3   ABLATION STUDY ON OT COEFFICIENT $\lambda$

We also study the effect of the hyperparameter $\lambda$ in the performance of $\alpha$-*rank* in exposure-based fairness. Specifically, we explore how varying values of $\lambda$ influence the quality of recommendations when $\alpha$ is set to different values within the range of 0 to 3.

Figure 13 presents that both Gini@10 and eCN@10 curves show a consistent decreasing trend as $\lambda$ increases. This trend becomes less pronounced as $\lambda$ reaches larger values. The experiment also verifies the parameter $\lambda$ in the OT projection controls the smoothness and the convexity of the exposure probability distribution between items.

Furthermore, it is noteworthy that curves for different $\alpha$ are basically the same. In the exposure-based $\alpha$-rank algorithm, the weight $\eta_i$ of items in equation 3 is the same for all items. In simpler terms, every item holds the same weight within the system, resulting in the same underlying principle regardless of the specific value of $\alpha$.

