# OpenReview forum: "α-Rank: Unified Item-Fair Ranking from A Cooperative Game Theory View"
_ICLR.cc/2024/Conference — ICLR 2024 Conference Withdrawn Submission_

### Official Review · Reviewer_3aBo · 2023-11-01

**Soundness:** 2 fair
**Presentation:** 1 poor
**Contribution:** 2 fair
**Rating:** 3
**Confidence:** 4

**Summary:**

This paper first gives a unified characterization of the ranking of items over buyers under different fairness metrics, i.e., utilitarianism, dealism and egalitarianism(the definition specified in the paper). Then, they introduced the $\alpha$-fairness (which is already defined in [1]) to unify these three types of fairness. Then, they obtain the final result by an OT projection process. This paper provides some visualizations of different Pareto frontiers, and compared their results with FairRec, FairRec++, Welf on advertising dataset.


[1]: Bertsimas, D., Farias, V. F., & Trichakis, N. (2012). On the efficiency-fairness trade-off. Management Science, 58(12), 2234-2250.

**Strengths:**

- The paper proposed to use OT for fair ranking, which is novel.
- The paper has many figures and tables. These figures are well tuned and informative.
- The paper has a pseudo code of their algorithm, and also an empirical evaluation of this algorithm.

**Weaknesses:**

- The use of word utilitarianism, dealism, and egalitarianism is a bit problematic:
-- This article employs obscure and convoluted vocabulary to redefine some properties that have already been well-defined in the fairness and economic literature, i.e., individual fairness, group fairness, and social welfare.
-- The reviewer has personally checked the literature, the word "egalitarianism" appeared **only once** in [4].

- Citation Issue:
-- some proper literature on optimal transport has not been cited, e.g., [2].
-- In the related work section, the **majority** of the works mentioned date back to **before 2000**, and this issue is particularly pronounced in the paragraph related to **cooperative games** and **fairness principles**. For instance, [3] should be included in the paragraph discussing cooperative games.

- Problem Formulation Issue:
-- $\mathbf{v}$ is defined both as a vector and a function of $\mathbf{x}_u$.
-- For the definition of $\mathcal{D}$, the correct formulation should be $\mathbf{1}^\top \mathbf{x}_u  \leq K$?
-- The original characterization of OT doesn't have the entropy term $H(x)$.


- Minor presentation issues:
-- two "in such games" in the intro.
-- In paragraph **item fairness methods**, the sentence "For different fairness aspects, Current mainstream", the "C" should be in lowercase.
-- In the second paragraph of introduction, the sentence "which equalizes the **utilities** of all item **utilities**", duplicate use of the word "utility".
-- It would be better to place the definitions of "CTR" and "CVR" in the caption of Table 1.
-- In paragraph **cooperative games**, the sentence "Another approach, proposed by (Nash Jr, 1950), is rooted in the concept of bargaining. " has an extra parathesis.
--The word charging can be replaced by other words, e.g., payment, bid, etc.

[2] Monge, G. (1781). Mémoire sur la théorie des déblais et des remblais. Mem. Math. Phys. Acad. Royale Sci., 666-704.

[3]  De Clippel, G., & Rozen, K. (2019). Fairness through the lens of cooperative game theory: An experimental approach. Available at SSRN 3527069.

[4] Bertsimas, D., Farias, V. F., & Trichakis, N. (2011). The price of fairness. Operations research, 59(1), 17-31.

**Questions:**

See weakness section.

---

### Official Review · Reviewer_9VZx · 2023-11-02

**Soundness:** 3 good
**Presentation:** 2 fair
**Contribution:** 3 good
**Rating:** 6
**Confidence:** 3

**Summary:**

The authors introduce a smoothed $\alpha$-rank framework to unify different fairness concepts like utilitarianism, dealism, and egalitarianism under a cooperative game framework, where items are akin to players dividing a "cake" of user attention. Experiments show that $\alpha$-rank outperforms baseline methods effectively.

**Strengths:**

- The paper introduces a unique perspective by framing item fairness in ranking systems within the context of cooperative game theory.
- The employment of optimal transport to conduct item fairness is a practical contribution.

**Weaknesses:**

Some notations are confusing, e.g., in equation (1) the **v** is a vector, then what does **v($x_u$)** mean?

**Questions:**

1. why the ranking problem can be formalized into equation (1)? and in equation (2), what is range of summation for $i$?

2. could you explain why the final ranking result can be obtained by the optimal transport projection?

3. could you compare your result Theorem 3 with Theorem 1 in [1]? I feel these two results bear similar message when $\alpha$ is zero or $\alpha$ is large.

[1] Yao, F. et al. (2023). Rethinking Incentives in Recommender Systems: Are Monotone Rewards Always Beneficial?. Advances in Neural Information Processing Systems, 37.

---

### Official Review · Reviewer_u8er · 2023-11-06

**Soundness:** 2 fair
**Presentation:** 1 poor
**Contribution:** 2 fair
**Rating:** 3
**Confidence:** 4

**Summary:**

This paper deals with the problem of recommending items to users in a fair manner, when each item derives different utilities from being recommended to each user, and each user can be recommended at most K items. The goal is to find a recommendation of items to users that is fair to the items. The authors consider three objectives or solution concepts: utility maximizing, Nash bargaining, and egalitarian (maximizing the utility of the worst off item) and propose a framework that allows to smoothly change the emphasis between these three objectives. The authors also provide a bound on the price of fairness and the compatibility of the different fairness notions with other desirable axioms.

**Strengths:**

- The main conceptual contribution of the paper is the extension of several notions of fairness and other axioms from the resource allocation literature to the recommender systems setting studied in the paper.
- These contributions appear to be novel in the related literature, which is surprising as they seem quite natural to consider. The topic seems relevant to ICLR.
- The main technical contribution is the formulation of the problem of finding a fair solution as an integer linear program. Since integer linear programs can be costly to solve, the authors provide a way to transform the solution of a variant of the program without integrality constraints which provides an upper bound on the value of the objective, and then constructing an integral solution from it using the Sinkhorn algorithm.

**Weaknesses:**

- It is not clear to me what fairness properties and axioms the final result of the \alpha-rank algorithm is guaranteed to satisfy. Could the authors please comment on this?
- I am not sure I understand the relationship between the value of the fairness objective of the output of \alpha-rank and that of the optimal solution to the original integer linear program in equation 2, i.e., between W(\alpha) and the fairness of L_K(u). Could you please help clarify? Is there a theoretical result I am missing? Alternatively, could you point to specific empirical results that illustrate this relationship? What are the axiomatic properties of the ranking?
- Is the objective to provide item fairness for each individual user decision or over all user decisions for fixed set of users? Could you please help clarify? The source of my confusion is the definitions of the fairness objectives starting at the end of Page 3 and the top of Page 4.
- The main theorems are stated too informally, and I do not understand what is being claimed just from reading them. It is possible to dig deeper and understand what is being proved by reading the proof, but it was not obvious from reading the theorem statement or the surrounding discussion. E.g. what does it mean for a fairness property X to adhere to an axiom Y? Do you mean that every feasible solution satisfying fairness property X (e.g. maximizing egalitarian objective) satisfies Axiom Y? Or does it mean that a feasible solution satisfy X and Y always exists? Could you please help clarify?
- Some technical terms used in the discussion need to be clarified in order to understand the paper properly. E.g. The authors state that the axioms in Section 4.1 show "how item fairness principles will behave when there are changes in available resources". There are several things that are not clear at this point. The term "resources" are never formally defined and it is not obvious from the discussion. I assume it refers to the number of items a user can be recommended. Is this right? Could you please clarify what is the difference between resources, attention, slots, cake, and other such terms used in the paper? How does Pareto optimality show how fairness principles behave as there are changes in available resources?
- There are several major typos and grammatical errors that make the paper hard to read.

Other comments placed here for the lack oxf another dedicated space:
- P1: Is "dealism" a commonly used term in the literature? Does this refer to the Nash bargaining solution?
- P1: ... papers advocate item from distinct principles ... -> item fairness
- P1: Please consider specifying what is an exposure slot.
- P1: utilitarianism objective ... maximizing summarization ... -> utilitarian objective ... maximizing summation?
- P1: dgalitarianism -> egalitarianism
- P2: we introduce the concept of \alpha-fairness to achieve a well balanced equilibrium among item fairness notions. What is meant by equilibrium here? Could you please elaborate?
- P3: vector x\in \mathbb{R}^{n\times m}. Do you mean matrix x? Is this a typo?
- P3: Please consider discussing what is a "resource" in Section 3.
- P3: It is not obvious from the text what v(x_u) or v_i(x_u) are. I assume v_i(x_u) is the utility for item from a decision x_u for user u. Is this right?
- P4, Axiom 5: The notion of a utility set has not been introduced if I remember correctly.
- In Fig. 1, what are the dashed lines?
- Overall, it might help to clarify whether resources refer to the K recommendation slots for each user. Are these indivisible or divisible? What does "cake" refer to?
- P6: each user can only be ranked among the top K items. Please consider rephrasing.

**Questions:**

Please see my questions in the comments above.

---

### Official Review · Reviewer_EN7p · 2023-11-06

**Soundness:** 3 good
**Presentation:** 2 fair
**Contribution:** 2 fair
**Rating:** 5
**Confidence:** 2

**Summary:**

The paper introduces a smooth α-fairness objective in order to unify different fairness concepts as a cooperative game problem. In these games, items are considered as the players dividing the “cake” of user attention. The paper analyzes the α-fairness objective from a theoretical way and introduces an efficient approach called α-rank. The paper also conducts experiments in two ranking applications: recommendation and advertising.

**Strengths:**

The paper faces a relevant and interesting problem, which is the one of providing a comprehensive theoretical framework encompassing several notions of fairness. It does so by using tools from cooperative game theory.

**Weaknesses:**

I have a major concern about the paper, which is how well it fits the ICLR community. Indeed, I expect that most of the researchers interested in ICLR are not familiar with basic notions of cooperative game theory. Thus, they would the paper difficult to read.

I also believe believe that the authors should make a better job selling the paper, making it clear from the beginning which are the possible applications of the framework and methods they provide in the paper.

**Questions:**

No questions.